# Analysis of Gait for Disease Stage in Patients with Parkinson’s Disease

**DOI:** 10.3390/ijerph18020720

**Published:** 2021-01-15

**Authors:** Mᵃ Helena Vila, Rocío Pérez, Irimia Mollinedo, José Mᵃ Cancela

**Affiliations:** Research Group HealthyFit, Institute of Health Research Galicia Sur (IISGS), Hospital University Complex of Pontevedra (CHOP), SERGAS, University of Vigo, 36310 Pontevedra, Spain; ghi22@uvigo.es (R.P.); irimia_mollinedo@hotmail.com (I.M.); chemacc@uvigo.es (J.M.C.)

**Keywords:** locomotion disorder, cadence, gait oscillation, speed of movement, neurodegenerative disease

## Abstract

Understanding the motor patterns underlying the movement of individuals with Parkinson’s disease (PD) is fundamental to the effective targeting of non-pharmacological therapies. This study aimed to analyze the gait pattern in relation to the evolutionary stages I–II and III–IV according to the Hoehn and Yahr (H&Y) scale in individuals affected by PD. The study was conducted with the participation of 37 PD patients with a mean age of 70.09 ± 9.53 years, and of whom 48.64% were women. The inclusion criteria were (1) to be diagnosed with PD; (2) to be in an evolutionary stage of the disease between I and IV: and (3) to be able to walk independently and without any assistance. Kinematic and spatial-temporal parameters of the gait were analyzed. The results showed differences in speed of movement, cadence, stride length, support duration, swing duration, step width, walking cycle duration, and double support time between the stages analyzed. These results confirmed the differences in PD gait pattern between stages I–II and III–IV. Different behaviors of the same variable were recorded depending on whether the right or left side was affected by PD.

## 1. Introduction

Parkinson’s disease (PD) is one of the most common age-related neurodegenerative diseases [1]. Progressive gait dysfunction is one of the primary motor symptoms in PD [2]. It is usually characterized by a reduction in step length and walking speed, and an increase in step time and cadence [3]. Disturbances in gait and posture are often resistant to drug treatment, deteriorate as the disease progresses, increase the likelihood of falls, and result in higher rates of hospitalization and mortality, thus having a negative impact on the patients’ quality of life [4]. The difficulty of walking within normal parameters is undoubtedly one of the greatest challenges faced by PD patients.

The Hoehn and Yahr (H&Y) scale is the most widely used instrument for establishing the degree of PD progression by simple staging [5,6]. It is used as the “gold standard” for checking other scales. According to the application of the H&Y scale, the most advanced stages of the disease lead to the worst quality of walking movements in PD patients, but limited information is available regarding the evolution of spatial-temporal and kinematic parameters, and the degree to which they contribute to this deterioration of the walking pattern. Therefore, further studies are needed in order to identify how to improve both the extent and quality of PD patients’ motor skills.

The analysis of walking patterns in healthy individuals has made it possible to obtain reference values for each variable (e.g., speed, cadence, stride length), which help to diagnose possible alterations in walking [7]. Movement disorders are differentially present throughout the development of various pathologies and may well reflect the underlying pattern of neurodegeneration [1,4]. It would be desirable to have the same values for pathologies that present characteristic patterns of walking, such as Parkinson’s disease (stiff gait), as these help in the early identification of pathologies that generate alterations in the gait pattern and cycle, if they have not already been diagnosed.

It is very important to establish which tools will help in the early identification of the different motor changes that occur during the evolution of the disease. From this point of view, it is necessary to make a detailed motor examination in order to determine as precisely as possible where and when it would be most advisable to intervene (from a motor point of view) throughout the course of the disease. Although instrumented motion analysis systems have been used for decades, their application has been mainly restricted to a laboratory environment. Today, with the technological advances in the analysis of movement in PD, new instruments are being presented for analysis, among which portable sensor technology [8], robotic rehabilitation [9], inertial sensors [10,11], and dynamometric or force platforms are notable [12].

There are numerous studies that analyze gait patterns in PD [13,14], but the number decreases when spatial-temporal and kinematic variables are analysed together [9,15]. There are several studies that reported differentiated symptoms in stages I–II and III–IV [5,6], but no research has been found that analyses and describes the evolution of spatial-temporal and kinematic variables in relation to the stage of PD progression and how this affects gait.

Therefore, there is a need to identify changes in the pattern of walking as a function of the stage of the disease by assessing kinematic and spatial-temporal parameters. This will enable health and fitness professionals and physiotherapists to design and implement customized exercise programs based on the specific needs of PD patients. Early identification is a key factor in establishing effective therapy and reducing costs in health and social care. For this reason, this study aimed to analyze the walking pattern (kinetic and spatial-temporal parameters) according to the evolutionary stage (I–II vs. III–IV) as specified by the Hoehn and Yahr (H&Y) scale in patients diagnosed with PD.

## 2. Materials and Methods

This is a primary study of a descriptive type (cross-sectional), in which a health-related population problem was analyzed; the study group comprised PD patients at different evolutionary stages of the disease. Thirty-seven PD patients aged between 61 and 87 years, with a mean age of 70.06 ± 9.53 years, participated in the study, of whom 18 were women. The inclusion criteria were being diagnosed with PD, the evolutionary stage of the disease being between I and IV according to the H&Y scale (a higher score indicates more severe impairment and disability), and being able to walk independently and without any assistance.

### 2.1. Subjects

The participants of the study were selected through a research proposal addressed to the Parkinson’s Association of the province of Pontevedra, specifically in the towns of Bueu and Villagarcia, by means of a collaborative framework agreement established between the association and the University of Vigo for research purposes. Participants volunteered to participate in the study, and those who met the inclusion criteria outlined above were selected. All participants were informed of the objectives of the study and signed the informed consent form. The study was approved by the ethics committee of the Regional Ministry of Health (code: CEIC: 2017/343) prior to the start of the research. All the procedures were undertaken in accordance with the ethical standards of the Helsinki Declaration of 1975, as revised in 2008, and Good Clinical Practice [16].

The sample size was calculated taking into account the systematic review of Parkinson’s Disease gait assessment using wearable motion sensors [17], with the location of the sensors on the lower back. For the calculation, the program G*power was used to estimate a proportion from an infinite total population (N), which maximizes the sample size. A confidence/safety level (1-alpha) of 95%, an accuracy (d) of 7%, and a ratio of 5% were chosen to maximize the sample size. Based on these data, the sample size (n) should be 37 subjects.

### 2.2. Instruments

In terms of anthropometric measurements, the height (cm) and weight (kg) of the participants were recorded with them being barefoot and wearing light clothing. The subjects’ body mass index (BMI) was calculated using the formula: weight/height^2^ (kg/m^2^). The measuring devices used were a Tanita TBF300 scale (TANITA Corporation, Tokyo, Japan) with an accuracy of 0.1 kg and a Handac stadiometer (Holtain Ltd., Crosswell, UK) with an accuracy of 1.0 mm. The anthropometric measurements were taken following the ISAK (International Working Group of Kinanthropometry) protocols [18].

A dynamometric corridor was used to evaluate the subjects’ gait by means of the pressure platforms E.P.S.-R1 of the LORAN-Engineering Company (LORAN-Engineering Company, Bologna, Italy) [12]. This corridor is composed of three platforms with 2304 sensors on an active surface of 2400 cm^2^, with a thickness of 7 mm, which facilitates the dynamic bipodal analysis of the patients. The kinetic variables evaluated by the dynamometric platforms in the gait corridor were as follows: average foot-support area (cm^2^), maximum foot pressure (Kpa), and average foot pressure (Kpa).

The inertial system used for the study of walking was Wiva^®^ Science [10]: an inertial sensor with the dimensions 40 × 45 × 20 mm and a weight of 35 g (Letsense Group, Bologna, Italy). The Wiva^®^ sensor includes an accelerometer, a magnetometer, and a gyroscope, which allows information to be recorded on the angular velocities reached by inertial detection devices placed on the L4-L5 spinal segment. In addition, Wiva^®^ collects data on the total time required to complete the task. All this information is sent via Bluetooth to a computer where it is stored using the Biomech Study 2011 v.1 software (Letsense Group, Bologna, Italy). The spatial-temporal parameters evaluated using Wiva^®^ Science were as follows: speed of movement (m/s), cadence (steps/min), stride length (m), stride length/height (m), step width (m), gait cycle time (s), time spent on double supports (s), average single support time (s), support time (% walking cycle), swing time (% gait cycle), right and left foot half angle (°), right and left leg acceleration gradient, right and left leg deceleration gradient, and step roll symmetry.

### 2.3. Procedure

Personal data was collected from each patient individually before the test was performed. Data collection was always carried out in the morning, 1.5 h after receiving medication, confirming that the patient was in an “ON” state. The data was recorded on 2 days a week (Tuesday and Thursday), and in two different locations, where the association has administrative and therapeutic offices.

For the collection of the walk analysis data, each subject was anonymously registered on the Biomech software, along with the following data: patient code, date of birth, gender, weight (kg), height (cm), shoe size, and anthropometric measurements of the upper and lower limbs.

The dynamometric corridor was made up of three platforms located on the floor and connected to each other by means of assembly elements that allowed them to be joined sequentially. These platforms were connected via a USB cable to the power supply and a laptop, which saved the data from the step-by-step analysis onto the Biomech program. In preparation for the test, the patients were requested to walk repeatedly on the platforms. Their walking gait was performed in the usual way, with a normal stride speed so as not to alter their gait pattern, while the repetitions and practice familiarized the patients with the texture and surface of the platforms. For the analysis of gait in the dynamometric corridor, the patients had to (1) stand at a distance of 1.5 m from the platform, and (2) undertake the following verbal instructions: “Walk at a normal speed until the mark located 1.5 m from the end of the corridor is reached.” The patients performed three runs along the corridor, and the most stable steps were selected.

The application protocol for the Wiva^®^ Science sensor was as follows. The Wiva sensor was fitted on each patient by means of an ergonomic waist band, at the height of the lumbar vertebra 5. Once the Wiva was placed on their waist, the patients were asked to walk on a straight line and follow any further instructions. The walking gait was to be performed in a normal and natural way. For the gait analysis, the patients had to (1) walk a distance of 3 m on a straight line with the Wiva^®^ Science sensor in place; (2) stop and remain immobile for 3 s; and (3) turn 360° and stop for another 3 s. The verbal instructions were as follows: “Walk at your normal speed until you reach the mark located 1.5 m from the end of the corridor” and “Stop, turn all the way around, and stop again”.

In both tests the gait cycle had to be performed in a natural way (with the usual speed and form), barefoot, without help, without load, and over three repetitions to achieve several analyses of the gait, whilst monitoring the support of both the right and the left foot. The patients had to listen carefully to the spoken instructions. Finally, readings were obtained from both feet, providing numerical and graphic data for each phase of the gait cycle to help in the interpretation of the results obtained in the study.

### 2.4. Statistical Analysis

A descriptive analysis was carried out using central tendency measures (mean and standard deviation/standard and percentages) of demographic, kinetic, and spatial-temporal variables, both globally and by segmenting the database by gender and stage of Parkinson’s disease (H&Y: I–II vs. III–IV). The Kolmogorov-Smirnov test (*p* > 0.05) was used to prove the normality of the variables under study. The homogeneity of the sample and the potential differences in variables of gender and stage of Parkinson’s disease were verified using Student’s t test for independent samples. With the aim of analyzing the possible association between the spatial-temporal parameters, a Pearson’s correlation analysis was performed to identify, if applicable, the degree of this association. The statistical analyses were carried out using the statistical package IBM SPSS v21 (IBM Corporation, Milwaukee, WI, USA) for Windows. Significance was considered for *p* < 0.05.

## 3. Results

Of the 37 participants in the study, 22 were in the H&Y I–II stage and 15 in the H&Y III–IV stage (Table 1). The youngest group of Parkinson’s patients presented a higher stage.

Table 2 presents the comparative analysis of spatial-temporal and kinetic parameters as a function of the PD stage. This analysis identified the presence of statistically significant differences between the groups in the main variables analysed.

Correlation analysis by stage (Table 3) for the kinetic variables revealed that the greatest number of variables with significant correlations were recorded for stage I–II patients. More specifically, the walking cycle duration variable was the variable showing the highest correlation with the other variables analyzed.

Figure 1 illustrates a decrease in walking speed concurrent with an increase in the stage of PD for both genders. The decrease is less significant in stage III–IV patients and in women compared with their respective counterparts.

In Figure 2, the cadence for both genders increases as the PD stage rises. The smallest increase was in women.

The behavior of the acceleration and deceleration gradient of the left leg in both stages was similar (Figure 3), but in the right leg there were significant differences in the deceleration gradient in stages H&Y I–II.

## 4. Discussion

The objective of the study was to analyze how the motor pattern of walking changes depending on the stage of PD. The findings of this study confirmed that in the analysis of the kinetic and spatial-temporal variables considered important in previous studies (speed of movement, cadence, duration of the walking cycle, time and duration of double support, duration of oscillation, step width, and stride length), there is a worsening as the disease progresses, with significant deterioration in stage III–IV compared with stage I–II. Gait speed was observed to differ according to gender. Women start out with worse values than men in stage I, but while their values stabilize throughout the course of the disease, men’s values worsen as the disease progresses. This detailed information suggests the need to improve the design of motor interventions aimed at slowing the progression of the disease as it relates to gait, both in terms of stage and gender.

As the disease progresses, changes in gait were accentuated between both groups, manifesting themselves through a decrease in the speed of movement. This behavior was maintained for each stage and gender, but there was a quantitative leap between stages II and III. Cadence increased as the stage increased, suggesting that an attempt was made to compensate for the loss of speed [19]. Speed and cadence behavior were consistent with other studies [13,20,21]. Other variables that influenced the gait pattern were an increase in the gait cycle duration, time and duration of double support, duration of swing, and step width. These variables reinforce the observation that walking speed decreases as the PD stage increases, and may be related to problems of dynamic balance [14,22]. The behavior of the variables above did not coincide with the study by Schlachetzki et al. [9], as far as PD stages I and III are compared.

Stride length behavior is worthy of separate treatment, as it generally decreased as PD progressed, coinciding with the results of Schlachetzki et al. [9]. In a more detailed analysis, an observation of right stride length showed opposite behavior to that of left stride length. The length of the right foot stride in stage I–II was lower than in stage III–IV. However, the decrease in left leg stride length from stage I and II to stages III and IV indicates further deterioration. Regarding the variables of a lateral analysis of PD patients, significant differences we also observed in the mean angulation of the right foot and the deceleration gradient of the right leg, these being significant differences that were not recorded in the left leg. We do not have an explanation for these results, which makes it necessary to analyze in greater depth the stride length variable. Some authors mentioned the asymmetry of motor dysfunctions in PD, with symptoms being more visible on the more affected side and deteriorating with the progression of the disease [15]. Others indicated that gait asymmetry is a relatively late change in gait dynamics [2]. The current study appears to be in line with Grajic et al. [23], stating that in early PD patients, gait parameters are asymmetric.

Focusing on the step width correlation analysis, significant correlations were demonstrated (with r higher than 0.6) between the duration of the walking cycle and time spent on double support in stages I–II and III–IV. The rest of the variables did not present this behavior. Stage I–II patients presented a greater number of relationships between variables, suggesting a more stable gait pattern. Siragy and Nantel [17] stated that each spatial parameter reflects a different aspect of motor control that contributes to a stable gait. Our study identified a large number of gait pattern variables that differed between stages I–II and III–IV in PD, which seems to indicate the need for different interventions depending on the stage of the patient, and even differing interventions for each leg.

To date, the physical therapies proven to be efficient adjuncts to medical treatment for people with PD are dance and water exercises [24], and there is evidence that robotic gait training is also effective [9,25]. More evidence is needed on the effectiveness of different therapies in phases III and IV with a specific focus on the different variables analysed and their evolution over the progression of PD.

This study had a number of limitations as follows. First, the sample was selected for suitability, and its size was small, which made it impossible for the study to analyze each stage, as stages I–II and III–IV had to be grouped together. Secondly, the evaluation tools had not previously been used by many researchers, which makes the discussion of the data more difficult. It is recommended that future research work is undertaken with a larger sample size, and that gait is analyzed at each stage of PD according to the Hoehn & Yahr classification reference system. It is also suggested that the Parkinsonian gait study be undertaken without the effect of medication (stage OFF), since our study was conducted while participants were following their usual medication cycles, and the effects of medication on gait cycle should be considered for analysis. Finally, the extent of the patients’ laterality and the areas that are affected should be incorporated into the analysis for a deeper investigation. Another limitation was the failure to identify the side most affected by PD.

To sum up, this study enabled us to provide additional information on the specific gait disturbances associated with each stage of PD, which is useful because a detailed gait analysis is key for understanding the complex pathophysiology of the disorder. Our results may also contribute to the development of a more objective evaluation of motor rehabilitation programs, the effectiveness of which will be demonstrated in the treatment of neurodegenerative disorders such as PD.

## 5. Conclusions

Regarding the kinetic and spatial-temporal variables that are considered important in the literature, this analysis revealed a significant deterioration linked to the progression of PD through its four stages. The most significant deterioration occurred in stages III–IV compared with stages I–II.

Regardless of the patients’ gender, this study identified a decrease in the speed of movement and an increase in the gait cadence between stages II and III.

Asymmetry in the spatial-temporal parameters of gait was recorded in the early stages of PD.

## Figures and Tables

**Figure 1 ijerph-18-00720-f001:**
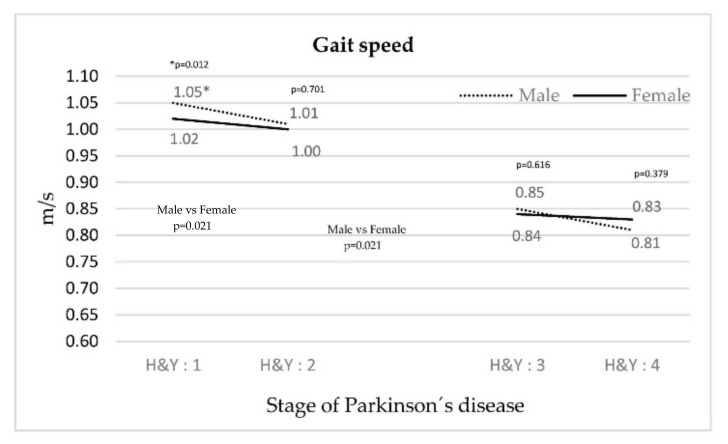
Evolution of gait speed depending on the stage of PD.

**Figure 2 ijerph-18-00720-f002:**
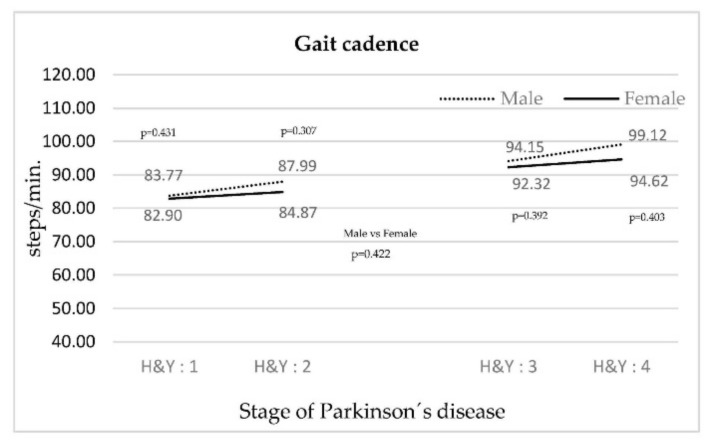
Evolution of gait cadence according to the stage of PD.

**Figure 3 ijerph-18-00720-f003:**
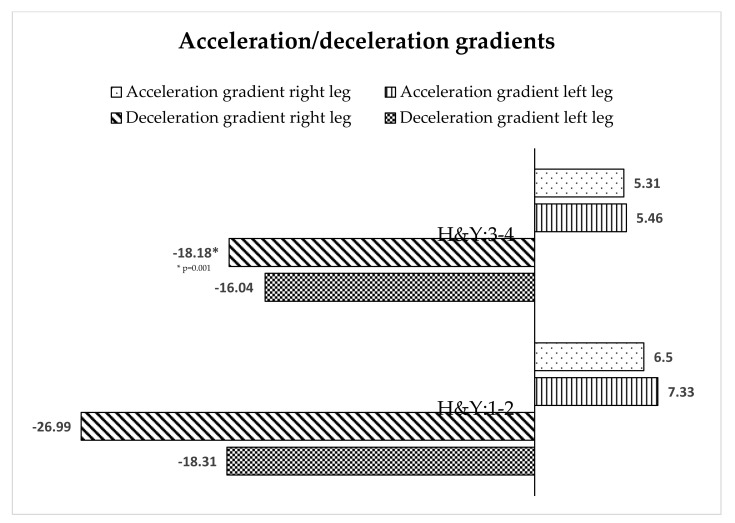
Acceleration/deceleration gradients of gait depending on the stage of PD.

**Table 1 ijerph-18-00720-t001:** Sociodemographic and clinical characteristics of the participants.

	Total *n* = 37	Hoehn-Yahr I–II (*n* = 22)	Hoehn-Yahr III–IV (*n* = 15)
	Mean ± SD	Mean ± SD	Mean ± SD
Age (years)	70.06 ± 9.53	70.45 ± 10.40	69.67 ± 8.41
Gender, male (%)	50.6%	54.5%	46.7%
Hoehn-Yahr stage	2.49 ± 0.72	1.86 ± 0.31	3.13 ± 0.41

**Table 2 ijerph-18-00720-t002:** Analysis of gait parameters (spatial-temporal and kinetic) in patients with Parkinson’s disease (PD).

	Hoehn-Yahr Stage				Student’s t	
	I–II		III–IV			
	Mean	SD	Mean	SD	t	*p*
Gait speed (m/s)	1.03	0.24	0.83	0.17	2.794	0.001
Cadence (steps/min)	84.80	13.77	95.08	10.55	2.089	0.001
Stride length (m)	1.08	0.22	1.02	0.21	1.012	0.031
Right step length (m)	0.49	0.21	0.53	0.22	0.918	0.039
Left step length (m)	0.59	0.23	0.49	0.20	1.134	0.001
Stride length/height (m)	0.65	0.11	0.64	0.13	0.086	0.932
Step width (m)	0.10	0.02	0.12	0.04	0.967	0.001
Stride time (s)	1.24	0.12	1.49	0.20	1.786	0.001
Double support time (s)	0.12	0.04	0.22	0.04	1.889	0.001
Single support time (s)	0.50	0.08	0.52	0.01	0.563	0.577
Right step time (s)	0.48	0.08	0.52	0.01	0.645	0.435
Left step time (s)	0.52	0.09	0.53	0.02	0.321	0.675
Stance (% gait cycle)	62.62	2.93	72.51	5.52	3.123	0.001
Swing duration (% gait cycle)	37.38	4.36	27.49	4.47	3.789	0.001
Right foot angulation (º)	9.00	8.58	11.66	10.88	3.001	0.001
Left foot angulation (º)	9.60	9.14	9.50	7.89	1.344	0.195
Acceleration gradient right leg	6.50	2.48	5.31	3.95	1.214	0.239
Acceleration gradient left leg	7.33	3.59	5.46	1.70	0.676	0.321
Deceleration gradient right leg	−26.99	18.12	−18.18	12.50	2.974	0.001
Deceleration gradient left leg	−18.31	14.14	−16.04	13.54	0.445	0.543
Swing symmetry in stride	1.00	0.18	1.04	0.26	0.387	0.712
Foot support surface (cm^2^)	122.85	39.46	126.00	24.63	0.768	0.376
Average foot pressure (Kpa)	147.8	24.72	123.69	47.85	0.967	0.167
Maximum foot pressure (Kpa)	400.79	73.89	414.89	125.05	1.101	0.109

**Table 3 ijerph-18-00720-t003:** Correlations between spatial-temporal parameters depending on the stage of the disease.

	Hoehn-Yahr Stage I–II				Hoehn-Yahr Stage III–IV			
	Gait Speed (m/s)	Cadence (Steps/min)	Stride Length (m)	Step Width (m)	Gait Speed (m/s)	Cadence (Steps/min)	Stride Length (m)	Step Width (m)
Stride time (s)	r = −0.644 sig = 0.012	r = 0.420 sig = 0.047	r = −0.663 sig = 0.021	r = 0.818 sig = 0.004	r = −0.331 sig = 0.048	r = 0.579 sig = 0.038	r = −0.438 sig = 0.041	r = 0.666 sig = 0.043
Double support time (s)	r = −0.471 sig = 0.017	r = 0.414 sig = 0.049	r = −0.244 sig = 0.041	r = 0.611 sig = 0.031	r = −0.201 sig = 0.056	r = 0.477 sig = 0.050	r = −0.388 sig = 0.054	r = 0.633 sig = 0.049
Swing duration (% gait cycle)	r = 0.414 sig = 0.044	r = −0.350 sig = 0.092	r = −0.601 sig = 0.011	r = −0.518 sig = 0.040	r = 0.701 sig = 0.145	r = −0.698 sig = 0.144	r = −0.655 sig = 0.129	r = −0.596 sig = 0.346
Deceleration gradient	r = 0.569 sig = 0.017	r = −0.470 sig = 0.022	r = 0.368 sig = 0.046	r = 0.345 sig = 0.045	r = 0.243 sig = 0.349	r = −0.369 sig = 0.222	r = 0.545 sig = 0.040	r = 0.505 sig = 0.043

## Data Availability

Data available on request due to restrictions eg privacy or ethical.

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
