# Peer review of "Analysis of Gait for Disease Stage in Patients with Parkinson’s Disease"

_ijerph, 2021, doi:10.3390/ijerph18020720_

Round 1

Reviewer 1 Report

The paper aimed to find out the relationship between various gait parameters with different stages of Parkinson's disease. The main conclusion is that different stages of the disease affects the patient's performance especially on the speed and cadence. However, to the reviewer, the findings aren't too surprise. Also, the numbers of patients involved are too few. It would be better, other than the two parameters discussed, to include more studies on how other parameters will be changed with the disease progression. 

Here are some minor issues in the paper.

1) Line 92, a typo on "cm2". 

2) What do "*" and "**" mean in Table 2?

3) In Table 2, why do some items Italic? Do they have special meanings?

4) Table 3 needs to be re-sized for readability.

Author Response

Response to Reviewer 1 Comments

Point 1. The paper aimed to find out the relationship between various gait parameters with different stages of Parkinson's disease. The main conclusion is that different stages of the disease affects the patient's performance especially on the speed and cadence. However, to the reviewer, the findings aren't too surprise. Also, the numbers of patients involved are too few. It would be better, other than the two parameters discussed, to include more studies on how other parameters will be changed with the disease progression. 

Thank you for your input. We have modified the first paragraph of the discussion, referring more specifically to the variables that showed differences by stage of PD in gait pattern. We also refer to the differences that were identified with respect to gender and the evolution of gait pattern (line 197 to 206). As regards the size of the sample, we have included a paragraph about the calculation of the sample size (see lines 89-94). In the discussion we have also included a paragraph about the guidance that these results may give with regard to the design of future physical interventions in PD (lines 238-241). We hope that we have given an adequate response to your accurate comments.

We hope you find the answer adequate and appropriate.

Here are some minor issues in the paper.

1) Line 92, a typo on "cm2". 

Absolutely in agreement, this has been corrected (line 105).

2) What do "*" and "**" mean in Table 2?

Thank you for your input. As another reviewer suggested including the p and v values in Table 2, the symbols are no longer used. The values to which they corresponded were *: sig< 0.05, ** sig<0.001.

3) In Table 2, why do some items Italic? Do they have special meanings?

We have changed the variables that were in italics to the same format as the rest of the variables. It was a formatting error, and these variables have no special significance.

4) Table 3 needs to be re-sized for readability.

 Thank you very much for the suggestion. We have changed the size of the content in table 3 for better Reading.

Reviewer 2 Report

This study aims to analyze the gait pattern in relation to the evolutionary stage between I-II and III-IV, according to the Hoehn and Yahr scale (H&Y) in people affected by PD. The study was conducted with the participation of 37 PD patients. The findings of this study confirm that in the analysis of the kinetic and spatial-temporal variables considered important in previous studies, there is a worsening as the disease progresses, with significant deterioration in stage III-IV as compared to stage I-II. 

The findings are in line with everyday clinical observation of the PD patients, but it seems not to add relevant information respecting to the state of the art. Please, improve the Introduction section by adding information on advanced studies in the field, that comprises the assessment of gait parameters in PD, by combining clinical tools and instrumental gait analysis assessment in innovative rehabilitation interventions (i.e. Bevilacqua et al. 2020. Rehabilitation of older people with Parkinson's disease: An innovative protocol for RCT study to evaluate the potential of robotic-based technologies BMC Neurology202020(1)186), in order to reinforce the soundness of the approach used in the paper.

However, the most negative issue is related to the small number of participants involved that does not allow any geralization of the findings. It is suggested to include more observations, as there is not intervention to conduct but only assessment, so it is strictly necessary to include a higher number of patients to derive any conclusion. 

Moreover, the Discussion/Conclusion section should be improved with information on specific interventions that may be beneficial after the findings, in order to give precise input to the health professionals on how the rehabilitation programs can be improved after the findings.

Small issues to correct:

  • In the Section 2.1 from line 74 to 77, pls correct the "XX":

"The participants of the study were selected through a research proposal addressed to the Parkinson's Association of the province of xx, specifically in the towns of xx and xx, by means of a collaborative framework agreement established between the association and the University of xx for research purposes"

  • In the references list, pls correct the double numbering 

Author Response

Response to Reviewer 2 Comments

Point 1. This study aims to analyze the gait pattern in relation to the evolutionary stage between I-II and III-IV, according to the Hoehn and Yahr scale (H&Y) in people affected by PD. The study was conducted with the participation of 37 PD patients. The findings of this study confirm that in the analysis of the kinetic and spatial-temporal variables considered important in previous studies, there is a worsening as the disease progresses, with significant deterioration in stage III-IV as compared to stage I-II. 

The findings are in line with everyday clinical observation of the PD patients, but it seems not to add relevant information respecting to the state of the art. Please, improve the Introduction section by adding information on advanced studies in the field, that comprises the assessment of gait parameters in PD, by combining clinical tools and instrumental gait analysis assessment in innovative rehabilitation interventions (i.e. Bevilacqua et al. 2020. Rehabilitation of older people with Parkinson's disease: An innovative protocol for RCT study to evaluate the potential of robotic-based technologies BMC Neurology, 2020, 20(1), 186), in order to reinforce the soundness of the approach used in the paper.

We appreciate the input, and have changed the paragraph which is now as follows (line 52-56)

Although instrumented motion analysis systems have been used for decades, their application has been mainly restricted to a laboratory environment. Today, with the advances that technologies offer us in the analysis of movement in PD, new instruments are being presented for analysis, among which portable sensor technology [8], robotic rehabilitation [9], inertial sensors [10,11] and dynamometric or force platforms are notable [12].

A new bibliographical reference has been included: Bevilacqua et al. 2020. Rehabilitation of older people with Parkinson's disease: An innovative protocol for RCT study to evaluate the potential of robotic-based technologies BMC Neurology, 2020, 20(1), 186.

Point 2. However, the most negative issue is related to the small number of participants involved that does not allow any geralization of the findings. It is suggested to include more observations, as there is not intervention to conduct but only assessment, so it is strictly necessary to include a higher number of patients to derive any conclusion. 

We appreciate the input. A paragraph of the sample size calculation has been included in the line 89 to 94. “The sample size was calculated taking into account the systematic review of Parkinson´s Disease gait assessment using wearable motion sensors [17] with the location of the sensors on the lower back.  For the calculation, the programme G*power was used to estimate a proportion from an infinite total population (N), which maximises the sample size. A confidence/safety level (1-Alpha) of 95%, an accuracy (d) of 7%, and a ratio of 5% were chosen to maximize the sample size. Based on these data, the sample size (n) should be 37 subjects.”

A new bibliographical reference has been included "Brognara, L., Palumbo, P., Grimm, B., & Palmerini, L. (2019). Assessing gait in Parkinson's disease using wearable motion sensors: a systematic review. Diseases, 7(1), 18."

Point 3. Moreover, the Discussion/Conclusion section should be improved with information on specific interventions that may be beneficial after the findings, in order to give precise input to the health professionals on how the rehabilitation programs can be improved after the findings.

The following text has been included in line 258 to 241. “To date, the physical therapies proven to be efficient adjuncts to medical treatment for people with PD are dance, water exercises (Alves Da Rocha et al. 2015) and there is evidence that robotic gait training is also effective (Bevilacqua et al. 2020; Alwardat et al, 2018). More evidence is needed on the effectiveness of different therapies in phases III and IV with a specific focus on the different variables analysed and their evolution over the progression of PD”.

We do not believe that we can go any further into this section, as this study is descriptive and aims to focus on the need to reorganise the different physical therapies according to the stages of PD evolution.

The following publications have been included in the bibliography:

Alves Da Rocha P, McClelland J, Morris ME. Complementary physical therapies for movement disorders in Parkinson's disease: a systematic review. Eur J Phys Rehabil Med. 2015 Dec;51(6):693-704. Epub 2015 Jul 3. PMID: 26138090.

Bevilacqua et al. 2020. Rehabilitation of older people with Parkinson's disease: An innovative protocol for RCT study to evaluate the potential of robotic-based technologies BMC Neurology, 2020, 20(1), 186.

Alwardat M, Etoom M, Al Dajah S, Schirinzi T, Di Lazzaro G, Sinibaldi Salimei P, Biagio Mercuri N, Pisani A. Effectiveness of robot-assisted gait training on motor impairments in people with Parkinson's disease: a systematic review and meta-analysis. Int J Rehabil Res. 2018 41(4):287-296. doi: 10.1097/MRR.0000000000000312. PMID: 30119060.

Small issues to correct:

  • In the Section 2.1 from line 74 to 77, pls correct the "XX":

The “XX” have been completed and corrected. The resulting corrected text is as follows: "The participants of the study were selected through a research proposal addressed to the Parkinson's Association of the province of Pontevedra, specifically in the towns of Bueu and Villagarcia, by means of a collaborative framework agreement established between the association and the University of Vigo for research purposes"

  • In the references list, pls correct the double numbering 

Thanks for the input. The double numbering has been corrected. Now there is only single enumeration.

Reviewer 3 Report

This study was to analyze the walking pattern (kinetic and spatial-temporal parameters) according to the evolutionary stage (I-II vs III-IV) as specified by the Hoehn and Yahr scale (H&Y) in patients diagnosed with PD. The results and theme of this article is quite interesting and I think that this paper is a well written paper. The biggest strength of this study is PD patients stage classification, however, there are major weaknesses. I have decided to major revision on the publication of the manuscript

Major concern

Introduction

  1. (line 40 ~ 56)

The author used spatiotemporal and kinematic variables for gait analysis. However, there are only a few explanations for these variables for patients with Parkinson´s Disease. As far as I know, there is a lot of research on the analysis of walking patterns in Parkinson's disease. Please describe the difference between this study and the previous study in relation to the independent variable, and explain why this study is needed. Authors state “Studies investigating gait pattern alterations in PD by stage are rare” So, I think the strength of this study is stage clasiffication. Please take this into account. Please explain in detail the above-mentioned contents instead of software and sensors.

Method

  1. The authors recruited 37 individuals with Hoehn-Yahr I-II and with Hoehn-Yahr III-IV. The statistical power of a study is heavily dependent on the number of subjects in each group, the major determinant is the size of the smallest group. It is very probable that the study does not have the statistical power to detect meaningful results. Although the sample size is described in the limitation section, I recommend that the investigators should suggest the result of a statistical power analysis, by calculating the sample size needed.
  2. The mean of age is 70, 69 in each group. However, authors state “Thirty-seven PD patients aged between 49 and 87 years”. Why is the recruitment range of subjects wide? Since gait patterns can be affected by age, this should be written as a limitation.
  3. In Table 2, what is foot angulation? The angle of the ankle joint or the angle of the foot segment? Please, describe a process or reference that calculated the variable.

Results

  1. Interestingly, authors expressed the variables of gait speed and gait cadence as figures. I think this will help readers understand. However, figure does not contain statistical results. Therefore, please write the statistical method and result in the statistical analysis and result. Additionally, if authors' intention is to analyze gender effects through figures, the statistical analysis method should be considered except t test.
  2. In Table 2, authors expressed mean and standard deviation value. However, since the author mentioned that they used t-test in the method section, t and p vlaue should also be written. No statistical interpretation is possible with the results of the manuscript.
  3. Additionally, authors used symbol such as “*” and “**”. Please, explain it to readers.

Discussion

  1. In first paragraph (line 183 ~ 185), the authors are highlighting the techniques used in this study, however, this study should focus on the "gait pattern" as stated in the title and purpose. Please summarize the results shown in the table and figure in first paragraph of discussion.

  1. In line 196 ~ 208, authors stated difference in right and left of stride length and angulation. Is there any evidence that PD only affects one side? If not, the authors will have to explain why they did not analyze the "dominant/non-dominant" or "non-affect side/affect side".

Minor concern

I recommend that this manuscript should be edited by an English professional editor for more readable. There are several grammatical errors.

Author Response

Response to Reviewer 3 Comments

Point 1. This study was to analyze the walking pattern (kinetic and spatial-temporal parameters) according to the evolutionary stage (I-II vs III-IV) as specified by the Hoehn and Yahr scale (H&Y) in patients diagnosed with PD. The results and theme of this article is quite interesting and I think that this paper is a well written paper. The biggest strength of this study is PD patients stage classification, however, there are major weaknesses. I have decided to major revision on the publication of the manuscript

Major concern

Introduction

  1. (line 40 ~ 56)

The author used spatiotemporal and kinematic variables for gait analysis. However, there are only a few explanations for these variables for patients with Parkinson´s Disease. As far as I know, there is a lot of research on the analysis of walking patterns in Parkinson's disease. Please describe the difference between this study and the previous study in relation to the independent variable, and explain why this study is needed. Authors state “Studies investigating gait pattern alterations in PD by stage are rare” So, I think the strength of this study is stage clasiffication. Please take this into account. Please explain in detail the above-mentioned contents instead of software and sensors.

We appreciate the input, and have added the following paragraph:

There are numerous studies that analyse gait patterns in PD [13,14], but the number decreases when spatial-temporal and kinematic variables are analysed together [9,15]. There are several studies that report differentiated symptoms in stages I-II and III-IV (5,6), but no research has been found that analyses and describes the evolution of spatio-temporal and kinematic variables in relation to the stage of PD progression and how this affects gait.

Pistacchi M, Gioulis M, Sanson F, De Giovannini E, Filippi G, Rossetto F, Zambito Marsala S. Gait analysis and clinical correlations in early Parkinson's disease. Funct Neurol. 2017 (1):28-34. doi: 10.11138/fneur/2017.32.1.028. PMID: 28380321; PMCID: PMC5505527.

Method

  1. The authors recruited 37 individuals with Hoehn-Yahr I-II and with Hoehn-Yahr III-IV. The statistical power of a study is heavily dependent on the number of subjects in each group, the major determinant is the size of the smallest group. It is very probable that the study does not have the statistical power to detect meaningful results. Although the sample size is described in the limitation section, I recommend that the investigators should suggest the result of a statistical power analysis, by calculating the sample size needed.

Thanks for the input, which is highly relevant. We have included in line 89 to 94 a paragraph of the sample size calculation, which indicates that the sample is adequate. The sample size was calculated taking into account the systematic review of Parkinson´s Disease gait assessment using wearable motion sensors [12] with the location of the sensors on the lower back.  For the calculation, the programme G*power was used to estimate a proportion from an infinite total population (N), which maximises the sample size. A confidence/safety level (1-Alpha) of 95%, an accuracy (d) of 7%, and a ratio of 5% were chosen to maximize the sample size. Based on these data, the sample size (n) should be 37 subjects."

A new bibliographical reference has been included "Brognara, L.; Palumbo, P.; Grimm, B.; Palmerini, L. (2019). Assessing gait in Parkinson's disease using wearable motion sensors: a systematic review. Diseases, 7(1), 18."

  1. The mean of age is 70, 69 in each group. However, authors state “Thirty-seven PD patients aged between 49 and 87 years”. Why is the recruitment range of subjects wide? Since gait patterns can be affected by age, this should be written as a limitation.

In fact the age was wrong due to a transcription error, instead of 49 it should be 61. It has been changed in line 74.

  1. In Table 2, what is foot angulation? The angle of the ankle joint or the angle of the foot segment? Please, describe a process or reference that calculated the variable.

The angulation of the foot is the angle between the line of the foot in support on the ground, and the line of the foot in displacement.  This variable is automatically calculated through an algorithm that the system calculates (Inertial Sensor Wiva®). We hope this answers your questions.

Results

  1. Interestingly, authors expressed the variables of gait speed and gait cadence as figures. I think this will help readers understand. However, figure does not contain statistical results. Therefore, please write the statistical method and result in the statistical analysis and result. Additionally, if authors' intention is to analyze gender effects through figures, the statistical analysis method should be considered except t test.

In complete agreement, the p values had not been indicated in the different graphs, so the p values have been included in all the graphs and the graphs have been changed in the document.

  1. In Table 2, authors expressed mean and standard deviation value. However, since the author mentioned that they used t-test in the method section, t and p vlaue should also be written. No statistical interpretation is possible with the results of the manuscript.

The t and p values have been included in table 2. The table has been changed.

  1. Additionally, authors used symbol such as “*” and “**”. Please, explain it to readers.

Thank you for your input. As another reviewer suggested including the p and v values in Table 2, the symbols are no longer used. The values to which they corresponded were *: sig< 0.05, ** sig<0.001.

Discussion

  1. In first paragraph (line 183 ~ 185), the authors are highlighting the techniques used in this study, however, this study should focus on the "gait pattern" as stated in the title and purpose. Please summarize the results shown in the table and figure in first paragraph of discussion.

Thank you very much for your input, we think it is correct. We have modified the paragraph as follows (line 197 to 206):

“The objective of the study was to analyze how the motor pattern of walking changes depending on the stage of the PD patient. The findings of this study confirm that in the analysis of the kinetic and spatial-temporal variables considered important in previous studies (speed of movement, cadence, duration of the walking cycle, time and duration of double support, duration of oscillation, step width, and stride length), there is a worsening as the disease progresses, with significant deterioration in stage III-IV as compared to stage I-II. Gait speed is observed to be different according to gender. Women start out with worse values than men in stage I, but while their values stabilise throughout the course of the disease, men's values worsen as the disease progresses. With this more detailed information, the need to improve the design of motor interventions aimed at slowing the progression of the disease as it relates to gait, both in terms of stage and gender, has been raised.”

  1. In line 196 ~ 208, authors stated difference in right and left of stride length and angulation. Is there any evidence that PD only affects one side? If not, the authors will have to explain why they did not analyze the "dominant/non-dominant" or "non-affect side/affect side".

Unfortunately, we do not have this information, it did not seem relevant to us when we collected the data, so we will include it in the limitations (line 252 and 253). Thank you for your input, which will be taken into account for future research.

Minor concern

I recommend that this manuscript should be edited by an English professional editor for more readable. There are several grammatical errors.

This manuscript has been revised by a professional English editor.

Round 2

Reviewer 1 Report

Thanks for the authors' response. I am fine with it. 

Reviewer 2 Report

Authors replied adequately to the criticisms found in the first review.

Reviewer 3 Report

The authors have addressed my suggestions. In my opinion, this work is suitable to be accepted by the journal.